# Preventative Effect of Angiotensin Receptor Blockers on Moderate-to-Severe Cerebral Vasospasm Among Patients Who Received Interventions for Aneurysmal Subarachnoid Hemorrhage

**DOI:** 10.3390/biomedicines13020442

**Published:** 2025-02-11

**Authors:** Si Un Lee, Hyoung Soo Byoun, Min Jai Cho, Jeong-Wook Lim, Chang Hyeun Kim, Jae-Seung Bang

**Affiliations:** 1Department of Neurosurgery, Seoul National University Bundang Hospital, Seoul National University College of Medicine, Seongnam-si 13620, Republic of Korea; nsmidget@gmail.com; 2Department of Neurosurgery, Chungbuk National University Hospital, Cheongju-si 28644, Republic of Korea; ulbo811211@naver.com; 3Department of Neurosurgery, Chungnam National University Sejong Hospital, Sejong-si 30099, Republic of Korea; jeongwook.lim@gmail.com; 4Department of Neurosurgery, Pusan National University Yangsan Hospital, Yangsan 50612, Republic of Korea; mole83@daum.net

**Keywords:** aneurysm, subarachnoid hemorrhage, angiotensin receptor blocker, vasospasm

## Abstract

**Objectives**: We conducted a retrospective study to investigate the effectiveness of angiotensin receptor blockers (ARBs) in preventing moderate-to-severe cerebral vasospasm, which may influence patient outcomes in cases of subarachnoid hemorrhage resulting from aneurysmal rupture. **Methods**: Between 2016 and 2020, we treated 210 patients with aneurysmal subarachnoid hemorrhage (aSAH) caused by a ruptured cerebral aneurysm. We obtained the clinical and radiological characteristics of patients through medical records and divided them into two groups: those who were administered ARBs (ARB group) and those who were not (no-ARB group). **Results**: A total of 181 patients aged 19 years or older with aSAH, without vascular abnormalities (including vascular malformations and moyamoya disease), were enrolled in this study. The age of the enrolled patients was 59.01 ± 12.98 (mean ± standard deviation), and the sex ratio of males to females was 66:115, with a higher proportion of females. The ARB group had 29 and the no-ARB group had 152 participants. The overall incidence of moderate-to-severe vasospasm was 33.7%. The incidence of moderate-to-severe vasospasm in each group was 13.8% (4 patients) and 37.5% (57 patients), respectively. The Fisher grade (III–IV) [odds ratio (OR) of 2.732 (95% confidence interval [CI]: 1.343–5.560; *p* = 0.006)] independently increases the risk of moderate-to-severe vasospasm, while older age [OR = 0.963; 95% CI: 0.938–0.989; *p* = 0.006] and ARB administration [OR = 0.246; 95% CI: 0.079–0.771; *p* = 0.016] independently decrease this risk. **Conclusions**: Despite the potential adverse impacts associated with hypotension, the administration of ARBs may provide therapeutic benefits in preventing moderate-to-severe vasospasm. A multicenter randomized double-blind controlled trial is needed to further investigate the efficacy and safety of ARBs in preventing moderate-to-severe vasospasm in aSAH patients who have undergone interventions and are experiencing acute hypertension.

## 1. Introduction

The incidence of cerebral vasospasm following subarachnoid hemorrhage due to aneurysmal rupture has been variably reported as 30% to 90%, depending on the diagnostic methods and criteria used [1,2,3]. Angiographic vasospasm occurs in up to 90% of aneurysmal subarachnoid hemorrhage (SAH) patients [2]. However, the prognosis can vary with the severity of angiographic vasospasm. It has been reported that the occurrence of delayed cerebral ischemia (DCI) or cerebral infarction varies depending on the grade of angiographic vasospasm. Moderate-to-severe cerebral vasospasm is associated with symptomatic vasospasm, DCI, and infarction, which can be related to poor neurological outcomes or prognosis [4,5].

A recent study has shown that oral administration of nimodipine, a calcium channel blocker, can decrease DCI, subsequently enhancing neurological outcomes without major side effects, and it is widely used for the prevention and treatment of cerebral vasospasm following aneurysmal SAH [6,7]. However, the mechanisms of its effect remain unclear [6,7]. Therefore, research into the mechanisms of cerebral vasospasm and more effective prevention and treatment strategies are being conducted. Several pathophysiological mechanisms of cerebral vasospasm, including prolonged arterial contraction, structural changes in the arterial wall, the breakdown of blood products, and inflammatory response, have been reported [1]. Upregulation or downregulation of the renin–angiotensin–aldosterone (RAA) system and the endothelin (ET) system, involved in the pathophysiological mechanisms, can be considered important mechanisms in the development of cerebral vasospasm [8,9,10,11]. Angiotensin receptor blockers (ARBs), a type of antihypertensive drug, are commonly used in conjunction with calcium channel blockers (CCBs) for patients with hypertension and have been reported to be effective in preventing cerebral vasospasm in several animal experimental studies and preclinical research [12]. The purpose of this study is to investigate the effectiveness of ARBs in preventing moderate-to-severe cerebral vasospasm, which may influence patient outcomes in cases of subarachnoid hemorrhage resulting from aneurysmal rupture.

## 2. Materials and Methods

### 2.1. Patient Collection and Baseline Characteristics

The present study was approved by the institutional review board, which waived the need for informed consent. Between 2016 and 2023, we conducted a retrospective analysis of the medical records of patients treated at our hospital for subarachnoid hemorrhage resulting from ruptured cerebral aneurysms. Excluded from the study cohort were patients who did not receive any interventions, those who succumbed within a week of treatment—thus precluding the assessment of cerebral vasospasm onset or progression—, individuals for whom a comprehensive evaluation for cerebral vasospasm was not performed, and patients with cerebrovascular anomalies like Moyamoya disease, which rendered the assessment of cerebral vasospasm infeasible.

We obtained the clinical and radiological characteristics of patients through medical records and divided them into two groups: those who were administered ARBs (ARB group) and those who were not (no-ARB group). The characteristics analyzed included age, sex, smoking history, hypertension, initial Glasgow Coma Scale (GCS) score, magnesium therapy, statin use, Hunt–Hess grade, Fisher grade, location of the aneurysm, M1/intenal carotid artery (ICA) ratio and M1 diameter (measured at admission and on day 7 post-onset of aSAH), mean blood velocity and peak blood velocity by transcranial Doppler (TCD) ultrasonography (measured at admission, on day 7 and 10 days post-treatment), presence or absence of cerebral vasospasm, performing angioplasty, modified Rankin Scale (mRS) score at admission (pre-procedure) and 3 months after onset of aSAH, and intervention modalities (clipping and coil embolization).

### 2.2. Patient Management

After treatment, all patients were admitted to the intensive care unit for management. The patient was placed in an isolated single room where monitoring and medication administration were conducted. If there was no improvement in the patient’s level of consciousness, intubation and mechanical ventilation were maintained. Considering the state of cerebral edema, intracranial pressure was managed with the administration of mannitol or hypertonic saline. Neurological examinations were performed hourly to detect any changes, particularly to monitor for rebleeding, worsening of acute hydrocephalus, or exacerbation of cerebral edema. If the neurological status improved, plans were made for extubation and transfer to a general ward. The principle of postoperative management was to monitor the occurrence of cerebral vasospasm while maintaining euvolemia and normotension. To achieve this, we used an ARB (losartan), CCB (nicardipine), and norepinephrine. Postoperatively, nimodipine was administered as a priority. For patients with a history of ARB administration prior to intervention, both nimodipine and an ARB were given as first-line treatments immediately after the procedure. If blood pressure was not adequately controlled with ARB, CCB was added. For patients without a history of ARB administration, only nimodipine was initially administered after the intervention. If blood pressure was not adequately controlled, the attending physician had the discretion to administer either an ARB or a CCB, or to prescribe both ARB and CCB together. We planned to administer an ARB for at least two weeks following the initiation of treatment. Postoperative systolic blood pressure was maintained within the range of 100–140 mmHg.

We considered a drop in systolic arterial blood pressure of more than 40 mmHg for over 2 h under euvolemic normotension, requiring the continuous use of norepinephrine, to be indicative of abnormal hypotension. Nimodipine was administered intravenously at a dosage of 20 mg/kg/h. Once the fasting period was concluded, the medication was switched to oral administration and continued for at least two weeks. Nimodipine was administered for two weeks following the occurrence of cerebral hemorrhage and was extended for an additional week based on the presence and severity of vasospasm, before which it was discontinued. In the event of abnormal hypotension, the ARB was discontinued first, and if there was no improvement, norepinephrine was administered to raise blood pressure. During the maintenance of euvolemic normotension, induced hypertension was initiated if symptoms due to cerebral vasospasm occurred. Norepinephrine was administered to achieve a targeted mean arterial blood pressure of 100–120 mm Hg during vasospasm treatment. Efforts were made to maintain a normal body temperature. To monitor for the occurrence of hydrocephalus and the status of bleeding, brain computed tomography (CT) scans were conducted every 7 days. On the seventh day following the hemorrhage, either magnetic resonance angiography (MRA) with time-of-flight (TOF) imaging or transfemoral cerebral angiography (TFCA) was performed to assess the presence of vasospasm and the state of treatment. An initial TCD ultrasonography was conducted within 3 days after the subarachnoid hemorrhage to evaluate vascular condition, and TCD ultrasonography was performed daily for two weeks, unless there were special events, to monitor the condition of the vessels. In cases of moderate or severe cerebral vasospasm or symptomatic vasospasm, chemical angioplasty was performed.

### 2.3. Definition of Moderate-to-Severe Vasospasm and DCI from Cerebral Vasospasm

All patients underwent a preoperative evaluation with TFCA to assess the initial state of the vessels. Baseline values measured by TCD ultrasonography were established within 3 days, and to track vascular status, TCD ultrasonography was performed over the course of 2 weeks post-intervention, with both the peak velocity and mean velocity of the middle cerebral artery (MCA) being monitored. The occurrence of cerebral vasospasm was investigated during two periods: around the seventh day the maximum values were observed, and around the tenth day the maximum values were noted. Moderate-to-severe vasospasm was identified based on a 150% increase from the baseline mean blood flow velocity or a 200% increase from the peak blood flow velocity measured by TCD ultrasonography. The M1/ICA ratio and degree of M1 segment stenosis calculated from an MRA time-of-flight image or TFCA performed on the seventh day post-intervention was used to determine the severity of the vasospasm; a vasospasm was defined as moderate-to-severe if there was a reduction of more than 33% in the ratio or if there was more than 33% stenosis in the M1 segment itself. On the seventh day after the procedure, even if no signs of vasospasm were observed on TCD monitoring, we routinely planned to perform MRA or TFCA to evaluate vascular condition. Additionally, if vasospasm was suspected on TCD, we conducted further TFCA to assess the vascular status and immediately performed chemical angioplasty if necessary for intervention. However, in cases where MRA or TFCA could not be performed due to the patient’s clinical condition or personal issues (particularly financial constraints), diagnosis had to rely solely on TCD. DCI is defined as the following: (1) symptomatic vasospasm, which includes clinical deterioration, a new focal deficit, decrease in level of consciousness, or both; and (2) infarction attributable to vasospasm, indicated by a new infarct visible on CT and/or MRI scans that was not present on the admission or immediate postoperative scan [4].

### 2.4. Primary and Secondary Endpoint

The primary endpoint of this study was to assess the incidence of moderate-to-severe vasospasm. Moderate-to-severe vasospasm was evaluated by dividing the patients into the ARB group and the no-ARB group. The secondary endpoint was to identify independent risk factors associated with moderate-to-severe vasospasm.

### 2.5. Statistical Analysis

All statistical analyses were conducted using IBM SPSS Statistics version 22.0 (IBM Corp., Armonk, New York, NY, USA). Continuous variables were reported as the mean ± standard deviation. The Chi-square test was utilized for categorical variables. Results were deemed statistically significant if the *p*-value was less than 0.05. Regression analysis was performed to adjust the numerical disparity in the two groups. The vasospasm-free survival rate according to the ARB administration was compared using the Kaplan–Meier method and statistical differences were revealed by log-rank test.

A univariate analysis was carried out to determine the association between risk factors and the occurrence of moderate-to-severe vasospasm. Following this, a multivariate logistic regression analysis was performed on factors that achieved a *p*-value of less than 0.2 in the univariate analysis to identify independent risk factors for moderate-to-severe vasospasm. A factor was considered an independent risk factor for moderate-to-severe vasospasm if it had a *p*-value of less than 0.05 in the multivariate analysis. The GPower 3.1 software is a commonly used power analysis program in the social sciences, utilized to calculate the effect size or the required sample size in statistical tests [13].

For a power analysis using linear multiple regression analysis, with an effect size of 0.15, an alpha of 0.05, a power of 0.80, and 11 predictors, the minimum required sample size is 123. Our sample size is 181, which is greater than the minimum sample size required for the power analysis. This meets Cohen’s (1988) standard of a power level of 0.8 or higher, indicating that the results of our study are statistically significant.

## 3. Results

A total of 210 patients were treated in our institution due to aneurysmal subarachnoid hemorrhage from 2016 to 2020. Among them, 11 patients were expired within a week due to initial poor clinical condition. Since one patient had moyamoya disease, we could not confirm whether the vasospasm was present during the hospital course. There were 17 patients for whom we could not estimate vascular condition due to a poor temporal window, which led to failure measuring initial or following values measured by TCD ultrasonography or the absence of subsequent radiologic evaluations. Finally, 181 patients enrolled in this study (Figure 1).

### 3.1. Baseline Characteristics

Of the 181 patients enrolled this study, 29 were administrated an ARB during the hospital course (ARB group) and the other 152 were not (no-ARB group). In the patient data, there were no statistical differences between the ARB and no-ARB groups, except for hypertension and statin use. In the radiologic data, only the proportion of patients with a higher Fisher grade at admission was greater in the ARB group than in the no-ARB group. Although the Fisher grade at admission was greater than in the ARB group, clinical outcomes according to the mRS score at discharge were better in the no-ARB group than in the ARB group (Table 1). In the ARB group, one patient, who had a preoperative mRS score of 2, died from rebleeding of coiled vertebral artery dissection aneurysm during rehabilitation. The clinical outcome of two patients deteriorated due to an unexplained increase in postoperative hematoma and procedure-related complications (cerebral infarction due to coil stretch).

### 3.2. Incidence of Moderate-to-Severe Vasospasm

The overall incidence of moderate-to-severe vasospasm was 33.7%. Fifty-seven patients (37.5%) suffered vasospasm in the no-ARB group. Among them, intra-arterial chemical angioplasty was performed in 25 (16.4%) patients. Four patients (13.8%) suffered vasospasm in the ARB group, and three patients (10.3%) were treated with angioplasty. The difference in the incidence of moderate-to-severe vasospasm between the two groups was statistically significant. DCI occurred in 19 patients (12.5%) in the no-ARB group and did not occur in the ARB group. This difference was statistically significant (Table 1). There was no statistically significant difference in the moderate-to-severe vasospasm-free survival rate between the no-ARB group and the ARB group (*p* = 0.699) (Figure 2).

Among 181 patients, 61 patients suffered vasospasm during the hospital course. There were no statistical differences between the non-vasospasm and vasospasm group for the initial M1/ICA ratio, initial mean blood flow velocity, and initial peak blood flow velocity measured by TCD ultrasonography. However, the M1/ICA ratio, mean blood flow velocity and peak blood flow velocity at 7 and 10 days after intervention were statistically different (Table 2).

### 3.3. Risk Factors for Moderate-to-Severe Vasospasm

Univariate analysis showed that older age and ARB administration are associated with vasospasm (*p* = 0.011 and *p* = 0.019). Multiple logistic regression analysis, using factors associated with vasospasm, showed that Fisher grade (III–IV) (OR = 2.732; 95%CI = 1.343–5.560; *p* = 0.006) independently increased the risk of vasospasm. And older age (odds ratio [OR] = 0.963; 95% CI = 0.938–0.989; *p* = 0.006) and ARB administration (OR = 0.246; 95%CI = 0.079–0.771; *p* = 0.016) independently decreased the risk for vasospasm (Table 3).

### 3.4. Case Illustration

A fifty-year-old female patient with a severe headache visited our emergency room. She had hypertension and had been taking a calcium channel blocker. Initial brain CT and CT angiography revealed an aneurysmal subarachnoid hemorrhage (Hunt–Hess grade 2 and modified Fisher grade 3) due to a ruptured left MCA bifurcation aneurysm (Figure 3A). An emergent TFCA was performed, revealing the aneurysm to have a maximal size of 6.21 mm and a neck size of 3.21 mm. Emergent coil embolization using the double microcatheter technique was successfully performed. The ruptured aneurysm occluded completely without procedural complications (Figure 3B). We inserted a lumbar drain to treat mild hydrocephalus, and the patient was admitted to the ICU for intracranial pressure control and vasospasm prevention. Initial TCD ultrasonography was conducted the day after coil embolization (Figure 3C,D). TCD ultrasonography monitoring continued for 2 weeks to detect vasospasm. The blood flow velocity of the left MCA was increased slowly. However, the patient experienced no clinical symptoms except for a headache. Seven days after coil embolization, she complained of severe headache and left-side visual disturbance. Immediate follow-up TCD ultrasonography revealed severe vasospasm, the mean blood flow velocity having doubled (Figure 3E,F). TFCA was conducted to confirm and treat the vasospasm. TFCA revealed severe vasospasm in the cerebral vessels of the left hemisphere (Figure 3G: the white arrow and circle denote regions of decreased blood flow attributable to vasospasm). Intra-arterial nimodipine angioplasty was performed for 4 days, after which the vasospasm and her symptoms gradually improved (Figure 3H–J: the white arrow and circle highlight areas of restored blood flow following intra-arterial nimodipine angioplasty). She was discharged at 3 weeks after coil embolization without neurologic symptoms. Her mRS score at 6 months post-embolization was 0.

## 4. Discussion

### 4.1. Diagnosic Modalities and Criteria of Cerebral Vasospasm

This study diagnosed vasospasm using TCD ultrasonography and angiographic imaging, including MRA with TOF imaging and TFCA. Various methods are employed to diagnose cerebral vasospasm, among which TCD ultrasonography, angiographic imaging including MRA, CT Angiography, TFCA, and dynamic perfusion CT are commonly utilized to date [4,5,14,15,16,17,18,19,20]. TCD ultrasonography, being non-invasive and repeatable, is considered a useful tool for the early detection of vasospasm, considering its accuracy [17]. Recent studies indicate that measuring blood flow velocity in the middle cerebral artery (MCA) via TCD ultrasonography is particularly effective in diagnosing vasospasm [15,18]. In a study evaluating the accuracy of TCD ultrasonography, it was reported that the mean blood flow velocity in the MCA exceeding 120 cm/s yielded a negative predictive value of 87%, while velocities greater than 200 cm/s demonstrated a positive predictive value of 87% [19]. Li et al. reported no significant difference in the incidence of vasospasm measured through both TCD ultrasonography and TFCA, with incidences of 87.78% and 83.33%, respectively [21].

In this study, the diagnostic criteria for moderate-to-severe vasospasm were based on a 150% increase from the baseline mean blood flow velocity or a 200% increase from the baseline peak blood flow velocity measured by TCD ultrasonography and more than a 33% constriction of the MCA as observed in angiographic image studies using MRA and TFCA. The diagnostic criteria for vasospasm vary among studies, but research employing angiographic imaging has diagnosed moderate-degree vasospasm with a 25% to 33% or more reduction in lumen diameter [4,5,17,22]. Studies using TCD ultrasonography have suspected vasospasm when the mean blood flow velocity exceeded 120 cm/s, diagnosing moderate degree vasospasm or higher when blood flow velocities were between 140 and 150 cm/s or above [15,21]. The mean blood flow velocity in TCD ultrasonography is calculated based on the formula MFV = (PSV + 2 × EDV)/3 (cm/s) [9], with maximum blood flow velocities of over 200 cm/s reported to be associated with cerebral ischemia and infarction [23]. Based on these findings, our study diagnosed moderate-to-severe vasospasm, which we consider to be substantiated by sufficient evidence.

As mentioned above, TCD, MRA, and TFCA are all essential methods for effectively diagnosing vasospasm. In this study, although the incidence of vasospasm was statistically lower in the ARB group, the TCD values did not show a significant statistical difference. This may be due to the overall low occurrence of vasospasm in both groups. To address this limitation, a large-scale prospective randomized study would be necessary. Nevertheless, this study is meaningful in that it evaluates moderate-to-severe vasospasm.

### 4.2. Incidence and Risk Factors of Cerebral Vasospasm

Cerebral vasospasm incidence is significantly elevated between 4 and 6 days post-subarachnoid hemorrhage (SAH), compared to the 1–3 days following the event, peaks at 7–9 days, and then diminishes at 10–12 days post-event [21]. Accordingly, this study analyzed the M1/ICA ratio and M1 diameter upon admission and on the seventh day post-treatment, as well as blood flow velocities measured by TCD ultrasonography upon admission, and on the seventh and tenth days post-treatment. Recent research suggests that cerebral vasospasm occurs in up to 90% of cases, with the incidence of moderate-to-severe vasospasm reported as between 31 and 45% [4,5,21,22,24,25]. These figures align closely with our study’s findings.

Reasonable risk factors for cerebral vasospasm include age, history of cigarette smoking, the severity of the SAH clot observed on CT, hypertension, and the clinical grade upon admission [2]. In our study on moderate-to-severe vasospasm, the Fisher grade (III–IV) independently increases the risk of moderate-to-severe vasospasm, while older age and ARB administration independently decrease this risk.

Alterations in the cerebral artery wall can be induced by age, lifestyle habits, and underlying health conditions. As age advances and with continuous smoking, alongside conditions like atherosclerosis and hypertension, vascular wall degeneration may occur, leading to changes in vascular wall compliance. Therefore, vascular wall degeneration can influence the occurrence of cerebral vasospasm and its treatment outcomes. Vasoconstriction risk may decrease in the presence of vascular degeneration, and the response to antivasospasm agents may also diminish in cases of vasospasm [2,26,27]. Our study uniquely found that age independently influenced the occurrence of vasospasm, with an increased age correlating with a reduced risk of moderate-to-severe cerebral vasospasm. A possible mechanism could be that, as brain atrophy progresses with age, the subarachnoid space widens, allowing a larger volume of blood to accumulate locally or flow into the CSF during the occurrence of SAH, thereby exacerbating the negative effects on vasospasm [2].

The severity of the SAH clot on CT is known in numerous studies as the most influential independent risk factor [1,2,27,28]. Cerebral vasospasm can be initiated and exacerbated by spasmogenic substances, one of the breakdown products of extravasated blood post-SAH [1]. In cases with a high Fisher grade (III–IV), indicating a substantial SAH clot, the risk of developing cerebral vasospasm increases, even with the use of antivasospasm agents, as does the risk for moderate-to-severe cerebral vasospasm.

In the RAA system and the ET system, the angiotensin II type 1 receptor and endothelin-1 are involved in mediating vasoconstriction, whereas the angiotensin II type 2 receptor is implicated in facilitating vasorelaxation. In the pathophysiological condition of SAH, elevated angiotensin II type 1 receptors can increase the risk of a long-lasting contraction induced by endothelin-1. ARBs, by intervening in this process, can suppress vasoconstriction, thus potentially preventing or aiding in the treatment of cerebral vasospasm [10,11,12,29,30]. An inflammatory response is one aspect of the pathophysiology of cerebral vasospasm. The PGF2α–prostacyclin–thromboxane system appears to play a role in the complex mechanisms leading to cerebral inflammation and vasospasm following subarachnoid hemorrhage (SAH), indicating its significance within the multifaceted process [31,32]. ARBs also seem to have a positive effect, not only for mitigating this inflammatory response, but also for preventing or reducing it [30,33]. Despite ARBs being antihypertensive agents with the effect of lowering systemic blood pressures, their use in the treatment of patients with aneurysmal SAH may have adverse effects. Nevertheless, their involvement in the modulation of vasoconstriction and inflammation warrants further exploration for potential therapeutic benefits in this context. Our study investigated a statistically significant lower incidence of moderate-to-severe cerebral vasospasm in patients administered ARBs, confirming that ARB intake is an independent factor in reducing the risk of vasospasm.

### 4.3. Limitation of the Study

This study has several limitations. First, this study may exhibit potential selection bias due to its retrospective nature. The patients included were treated at a single institution, employing a nearly identical surgical procedure. However, the values measured by TCD ultrasonography during the hospital course could be influenced by the patient’s condition, including hyperemia and unstable vital status. Second, the angiographic evaluation conducted on the seventh day of hospitalization may not have been entirely accurate in some respects due to the lack of uniformity in the method used, as it was performed using either MRA or TFCA, rather than a single standardized approach. Third, while the preventive effect of ARBs on moderate-to-severe vasospasm was observed, the group not administered ARBs surprisingly showed better clinical outcomes. This discrepancy could be attributed to the relatively smaller size of the ARB group, in which a higher rate of unexpected complications during the treatment process might have skewed the accurate assessment of clinical outcomes, as represented by the Modified Rankin Scale (mRS) score. Lastly, due to the nature of the retrospective study, ARB administration was not randomly assigned.

## 5. Conclusions

This study highlights that, in the initial management of SAH patients, particular attention should be given to age and volume of hemorrhage due to their association with the development of moderate-to-severe cerebral vasospasm. Despite the potential adverse impacts associated with hypotension, the administration of ARBs may offer therapeutic benefits in treating SAH patients, owing to their capacity to prevent moderate-to-severe cerebral vasospasm through diverse mechanisms.

## Figures and Tables

**Figure 1 biomedicines-13-00442-f001:**
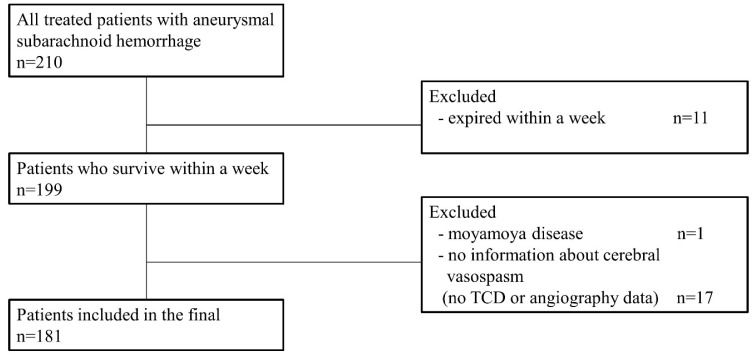
Flow chart of study participant selection.

**Figure 2 biomedicines-13-00442-f002:**
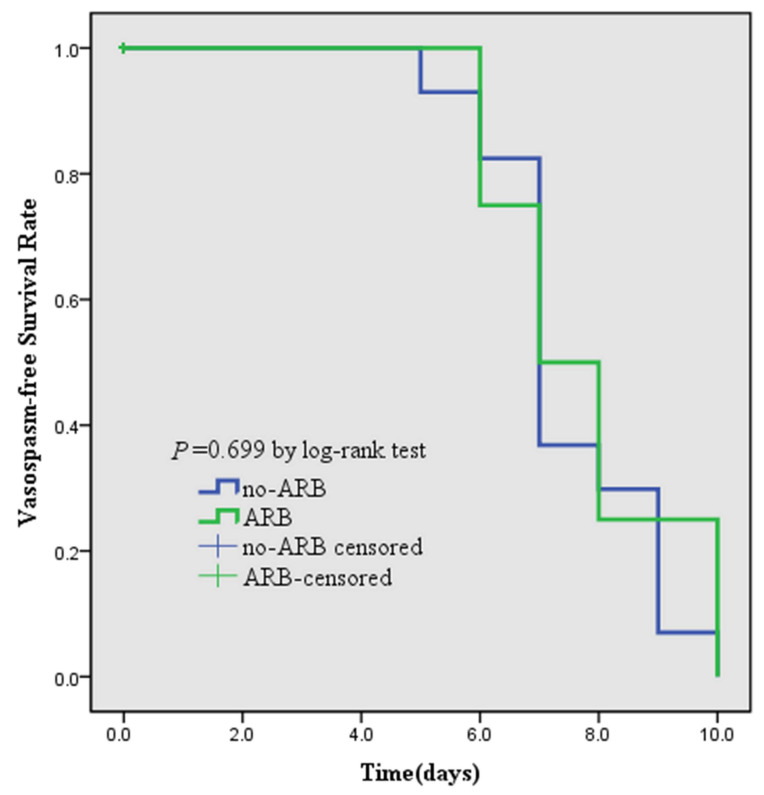
Kaplan–Meier curve depicting the vasospasm-free survival rate based on ARB administration.

**Figure 3 biomedicines-13-00442-f003:**
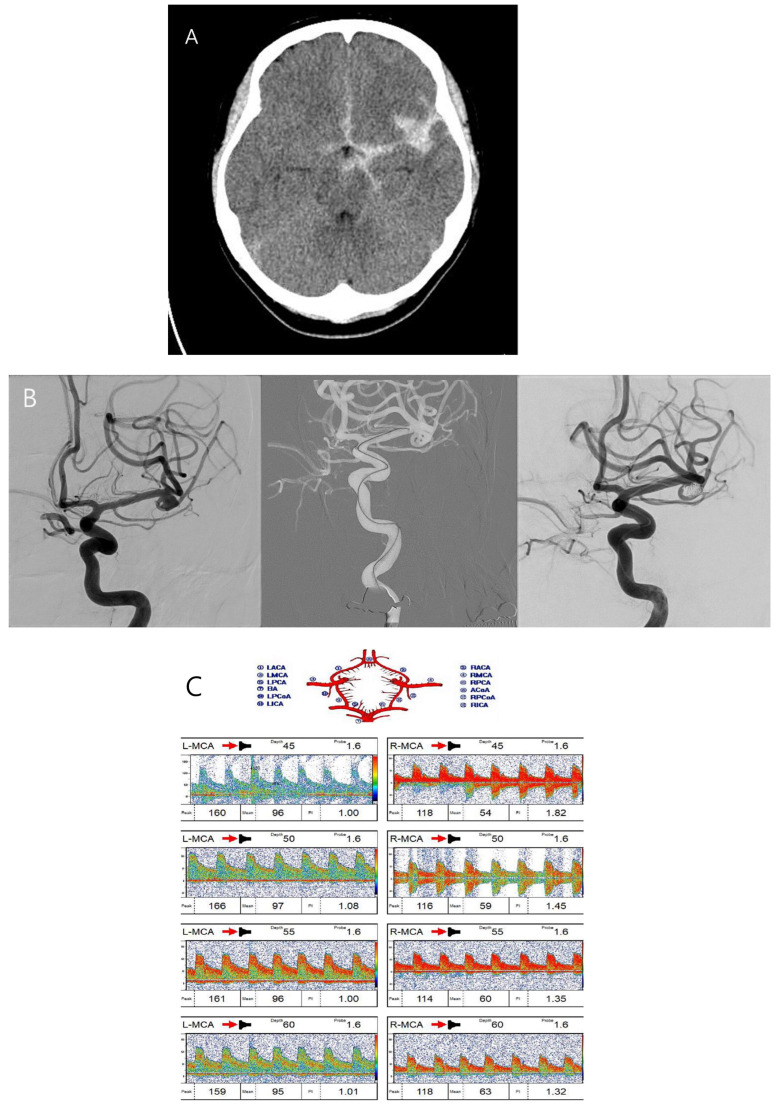
Case illustration of patient who suffered subarachnoid hemorrhage.

**Table 1 biomedicines-13-00442-t001:** Baseline characteristics.

Variables	No ARB (n = 152)	ARB (n = 29)	OR (95% CI)	*p*-Value
Age (mean ± SD), years	58.19 ± 13.136	63.31 ± 11.430	1.032 (0.999–1.066)	0.054
Sex (female), n (%)	96 (63.2)	19 (65.5)	0.902 (0.392–2.077)	0.809
Smoking, n (%)	48 (31.6)	8 (27.6)	1.212 (0.501–2.930)	0.670
HTN, n (%)	25 (16.4)	29 (100)	N/A *	<0.001
Initial GCS < 8	17 (11.2)	7 (24.1)	2.527 (0.940–6.792)	0.066
Treated with magnesium, n (%)	147 (96.7)	29 (100.0)	N/A	1.000
Statin use, n (%)	23 (15.1)	9 (31.0)	0.396 (0.161–0.978)	0.045
HH grade (≥3), n (%)	67 (44.1)	13 (44.8)	0.970 (0.436–2.157)	0.941
Fisher grade (≥3), n (%)	85 (55.9)	22 (75.9)	0.404 (0.163–1.002)	0.050
Location of aneurysm, n (%)			0.302 (0.082–1.107)	0.071
Ant. Circulation	145 (95.4)	25 (86.2)		
Post. Circulation	7 (4.6)	4 (13.8)		
M1/ICA ratio				
Initial	0.547 ± 0.110	0.524 ± 0.115	0.161 (0.006–4.530)	0.283
At POD #7	0.492 ± 0.129	0.485 ± 0.120	0.166 (0.073–2.978)	0.419
TCD Mean Velocity				
Initial	41.578 ± 27.611	39.43 ± 26.536	0.997 (0.983–1.011)	0.683
At POD #7	71.148 ± 37.915	63.198 ± 23.829	0.997 (0.988–1.006)	0.544
At POD #10	75.854 ± 38.015	61.769 ± 27.525	0.993 (0.984–1.002)	0.145
TCD Peak velocity				
Initial	62.155 ± 44.418	60.596 ± 44.866	0.999 (0.991–1.008)	0.855
At POD #7	105.089 ± 55.234	90.986 ± 37.729	0.998 (0.991–1.004)	0.457
At POD #10	106.526 ± 53.002	88.299 ± 41.728	0.996 (0.989–1.002)	0.183
Vasospasm, n (%)	57 (37.5)	4 (13.8)	3.750 (1.242–11.326)	0.019
Angioplasty, n (%)	25 (16.4)	3(10.3)	1.706 (0.479–6.073)	0.410
DCI, n (%)	19 (12.5)	0 (0)	N/A *	0.047
Pre-mRS (≤2), n (%)	83 (54.6)	14 (48.3)	1.289 (0.582–2.855)	0.532
mRS at 3 months (≤2), n (%)	117 (77.0)	16 (55.2)	2.716 (1.192–6.189)	0.017
Surgical clipping, n (%)	15 (9.9)	1 (3.4)	3.066 (0.389–24.166)	0.288
Coil embolization, n (%)	137 (90.1)	28 (96.6)	3.066 (0.389–24.166)	0.288

ARB: angiotensin receptor blocker, SD: standard deviation, HTN: hypertension, GCS: Glasgow Coma Scale, HH: Hunt–Hess, Ant.: anterior, Post.: Posterior, ICA: internal carotid artery, POD: postoperative day, TCD: transcranial Doppler ultrasonography, DCI: delayed cerebral ischemia, mRS: Modified Rankin Scale. *: the number of items is either all or none in the patient group, statistical values cannot be calculated.

**Table 2 biomedicines-13-00442-t002:** Differences in M1/ICA ratio and TCD velocities between non-vasospasm and vasospasm groups.

Variables	Non-Vasospasm (n = 120)	Vasospasm (n = 61)	*p*-Value
M1/ICA ratio (mean ± SD)			
Initial	0.546 ± 0.0963	0.539 ± 0.116	0.655
At POD #7	0.525 ± 0.121	0.302 ± 0.154	<0.001
TCD Mean velocity (mean ± SD)			
Initial	51.848 ± 18.974	50.276 ± 23.110	0.680
At POD #7	58.286 ± 18.153	187.477 ± 46.699	<0.001
At POD #10	58.672 ± 21.888	191.572 ± 43.621	<0.001
TCD Peak velocity (mean ± SD)			
Initial	79.315 ± 34.577	70.563 ± 37.853	0.185
At POD #7	89.466 ± 37.760	222.321 ± 67.201	0.002
At POD #10	86.466 ± 37.760	225.094 ± 61.352	<0.001

ICA: internal carotid artery, SD: standard deviation, POD: postoperative day, TCD: transcranial Doppler ultrasonography.

**Table 3 biomedicines-13-00442-t003:** Univariate and multivariate analyses of the risk factors for cerebral vasospasm.

Factors	Univariate	*p*-Value	Multivariate	*p*-Value
	Odd Ratio (95% CI)		Odd Ratio (95% CI)	
Age	0.968 (0.944–0.993)	0.011	0.963 (0.938–0.989)	0.006
Sex	0.923 (0.487–1.747)	0.805		
Smoking	1.784 (0.928–3.430)	0.083	1.552 (0.773–3.116)	0.217
HTN	0.522 (0.254–1.072)	0.076	1.322 (0.473–3.699)	0.595
Magnesium	0.756 (0.123–4.651)	0.763		
Statin use	0.873 (0.384–1.984)	0.747		
ARB	0.267 (0.088–0.805)	0.019	0.246 (0.079–0.771)	0.016
Additional CCB	0.394 (0.153–1.018)	0.054	0.515 (0.189–1.405)	0.195
Fisher grade (III–IV)	1.870 (0.976–3.583)	0.059	2.732 (1.343–5.560)	0.006
HH grade	1.497 (0.805–2.783)	0.202		
Tx. Modality (clip)	1.130 (0.374–3.413)	0.828		

HTN: hypertension, ARB: angiotensin receptor blocker, CCB: calcium-channel blocker, HH: Hunt–Hess, Tx.: treatment.

## Data Availability

Data are available upon reasonable request from the corresponding authors.

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
