# Peer review of "Preventative Effect of Angiotensin Receptor Blockers on Moderate-to-Severe Cerebral Vasospasm Among Patients Who Received Interventions for Aneurysmal Subarachnoid Hemorrhage"

_biomedicines, 2025, doi:10.3390/biomedicines13020442_

Round 1

Reviewer 1 Report

Comments and Suggestions for Authors

This paper reports on a single-centre retrospective cohort study of the frequency of moderate to severe vasospasm among patients with aneurysmal subarachnoid haemorrhage (aSAH) depending on whether they had received angiotensin-receptor blocker (ARB). In the study of 181 included patients, vasospasm was not-expectedly associated with higher Fisher grade, but surprisingly with lower age and, the aim of the study, use of ARB. The authors concluded that despite the potential adverse impacts associated with hypotension, the administration of ARBs may provide therapeutic benefits in preventing moderate to severe vasospasm.

There are a number of issues the authors may wish to address, some major. The most critical ones are:

1.      That ARB and if needed added CCB were used to treat high blood pressure among aSAH patients who had undergone interventional treatment. Also noepinephrine was used to raise BP if hypotensive. Thus, is this study really about needing vs not needing medications to maintain BP control within a certain range to ‘prevent’ vasospasm (I note the absence of evaluating the role of norepinephrine in Table 3)? Blood pressure rises after SAH - those with higher blood pressure are at higher risk of vasospasm (Faust K, Horn P, Schneider UC, Vajkoczy P. Blood pressure changes after aneurysmal subarachnoid hemorrhage and their relationship to cerebral vasospasm and clinical outcome. Clin Neurol Neurosurg. 2014 Oct;125:36-40. doi: 10.1016/j.clineuro.2014.06.023. Epub 2014 Jul 9. PMID: 25083804; Hosmann A, Klenk S, Wang WT, Koren J, Sljivic S, Reinprecht A. Endogenous arterial blood pressure increase after aneurysmal subarachnoid hemorrhage. Clin Neurol Neurosurg. 2020 Mar;190:105639. doi:10.1016/j.clineuro.2019.105639. Epub 2019 Dec 16. PMID: 31874423.). It is the 2nd of the bimodal peak of BP rise after SAH that is associated with severe vasospasm (Fontana J, Scharf J, Weiß C, Schmieder K, Barth M. The spontaneous arterial blood pressure rise after aneurysmal subarachnoid hemorrhage - a biphasic phenomenon. Clin Neurol Neurosurg. 2015 Oct;137:22-7. doi:10.1016/j.clineuro.2015.06.014. Epub 2015 Jun 19. PMID: 26123527.)

2.      CCB was added to ARB if needed – thus this is ARB-based therapy and not ARB alone or CCB alone, and ARB+CCB comprised a small proportion

3.      It is unclear if ARB(+/-CCB) were given before or after onset of vasospasm, before or after intervention – it appears to be ‘anytime’

4.      The achieved BP was not mentioned

5.      This was not a randomised controlled trial, but rather an observational study

6.      The way vasospasm was diagnosed is in doubt

7.      That functional outcome (which is the end product of all processes including vasospam) was WORSE in the ARB group and this was glossed over by the authors

The other issues:

1.      Line 2 – ‘for’ may be better replaced with ‘of’

2.      Line 3 - to replace ‘after’ with ‘among patients who received interventions for’ - to better reflect the inclusion criteria

3.      Line 16 – why is ‘verify’ used? Are there prior reports of successful ARB use in humans with aSAH that need verification? Maybe say ‘investigate’ if there are no prior reports

4.      Line 18 – Methods – briefly mention inclusion criteria

5.      Line 19 – clarify who these 210 patients are eg ‘with aSAH’

6.      Line 21 – Results - mention patient demographics and relevant characteristics

7.      Lins 24-27 – better to mention the direction of the association eg increased or decreased

8.      Lines 29-31- may be better replaced as they do not attend to the study question. Eg say “Multicentre randomised blinded controlled trails are needed to further study the efficacy and safety of ARBs for the prevention of moderate to severe vasospasm among aSAH patients who had undergone intervention and had acute high blood pressure’

9.      Line 42 – add ‘poor’ before ‘neurological’ to show the direction of the relationship

10.  Line 46 – I understand nimodipine is used to reduce the risk of DCI and it has little material effect on the prevention and/or treatment of vasospasm – please quote the relevant reference if otherwise

11.  Line 58 – ref 11 has been misquoted – I ask the authors to carefully read that paper

12.  Line 57 (major) – if CCBs and ARBs have actually been shown to be effective, what is the knowledge gap this study is trying to fill?

13.  Line 59 – is the aim to ‘verify’ what is already known? (See 2. above) What would be the added value of this study?

14.  Line 75 – please clarify that the ARB was received at any time during the admission (ie it can be before or after intervention, vasospasm)

15.  Lines 80 and 81 (major) – ‘post-treatment’ - is it ‘post-intervention’? ‘Post-admission? Or ‘post-onset of aSAH’ – though it is likely patients will be admitted on day of onset due to severe headache/loss of consciousness? Note that timing of vasospasms is usually based on day of onset

16.  Line 83 – ‘admission’ – do the authors mean pre-morbid? ‘discharge’ - mRS at 3 months post-aSAH may be better measure of functional outcome than at discharge

17.  Lines 86, 128, 135 – and throughout the text where relevant (major) - ‘treatment may be better replaced with ‘intervention’

18.  Line 96 – what is the target BP range for ‘normotension’? What ARBs and CCBs were used?

19.  Section 2.3 and 4.1 (very major) – it is unclear what was used to diagnose moderate to severe vasospasm? TCD, MRA? TFCA? Any of them? If any, the authors need to show that all the 3 methods used for diagnosing vasospam are highly correlated

20.  Line 131 (major) – I have concerned that only velocities were used to diagnose vasospasm when the recommendation is to use the Liebegard (MVA/tICA) ratio as well (Alexandrov AV, Sloan MA, Wong LK, Douville C, Razumovsky AY, Koroshetz WJ, Kaps M, Tegeler CH; American Society of Neuroimaging Practice Guidelines Committee. Practice standards for transcranial Doppler ultrasound: part I—test performance. J Neuroimaging. 2007 Jan;17(1):11-8. doi:10.1111/j.1552-6569.2006.00088.x. PMID: 17238867.)

21.  Lines 136-137 (major) – what is the authoritative reference for this angiographic criteria?

22.  Line 137 - what is the authoritative reference for this DCI criteria?

23.  Table 1 – please check the p-value for HTN, unlikely to be 0.941. M1/ICA ratio on which test? What is ‘pre-mRS’ – pre-morbid? On admission pre-intervention? If there was less vasospasm in ARB group, why are the TCD velocities not different – even more relevant issue as table 2 shows differences(major)

24.   Line 200 - is ‘operation’ different from ‘intervention’?

25.  Table 2 – do not use ‘p=0.000’, instead provide a value eg <0.001 if that is correct

26.  Results (major) – 4.1 - this should be spent discussing the results of their internal comparison of 3 methods the authors used, and the lack of congruity as mentioned in 23. Above

27.  Table 3 - what about the detrimental effects of norepinephrine on vasospasm? (Zeiler FA, Silvaggio J, Kaufmann AM, Gillman LM, West M. Norepinephrine as a potential aggravator of symptomatic cerebral vasospasm: two cases and argument for milrinone therapy. Case Rep Crit Care. 2014;2014:630970. doi:10.1155/2014/630970. Epub 2014 Nov 9. PMID: 25431686; PMCID: PMC4241707.)

28.  Line 294-296 – the direction of the association should be stated

29.  Line 305 (major) – a detailed explanation of this finding needs to be done as it is counter to the literature

30.  Line 327 – again, ‘verify’ may not be appropriate

31.  Limitations – the authors should add that the allocation d ARB was not randomised

Author Response

Response to Reviewer’s comment

Reviewer #1.

This paper reports on a single-centre retrospective cohort study of the frequency of moderate to severe vasospasm among patients with aneurysmal subarachnoid haemorrhage (aSAH) depending on whether they had received angiotensin-receptor blocker (ARB). In the study of 181 included patients, vasospasm was not-expectedly associated with higher Fisher grade, but surprisingly with lower age and, the aim of the study, use of ARB. The authors concluded that despite the potential adverse impacts associated with hypotension, the administration of ARBs may provide therapeutic benefits in preventing moderate to severe vasospasm.

There are a number of issues the authors may wish to address, some major.

Response: Thank you very much for taking the time to review this manuscript. Please find the detailed responses below and the corresponding revisions/corrections highlighted in the re-submitted files.

The most critical ones are:

  1. That ARB and if needed added CCB were used to treat high blood pressure among aSAH patients who had undergone interventional treatment. Also noepinephrine was used to raise BP if hypotensive. Thus, is this study really about needing vs not needing medications to maintain BP control within a certain range to ‘prevent’ vasospasm (I note the absence of evaluating the role of norepinephrine in Table 3)? Blood pressure rises after SAH - those with higher blood pressure are at higher risk of vasospasm (Faust K, Horn P, Schneider UC, Vajkoczy P. Blood pressure changes after aneurysmal subarachnoid hemorrhage and their relationship to cerebral vasospasm and clinical outcome. Clin Neurol Neurosurg. 2014 Oct;125:36-40. doi: 10.1016/j.clineuro.2014.06.023. Epub 2014 Jul 9. PMID: 25083804; Hosmann A, Klenk S, Wang WT, Koren J, Sljivic S, Reinprecht A. Endogenous arterial blood pressure increase after aneurysmal subarachnoid hemorrhage. Clin Neurol Neurosurg. 2020 Mar;190:105639. doi:10.1016/j.clineuro.2019.105639. Epub 2019 Dec 16. PMID: 31874423.). It is the 2ndof the bimodal peak of BP rise after SAH that is associated with severe vasospasm (Fontana J, Scharf J, Weiß C, Schmieder K, Barth M. The spontaneous arterial blood pressure rise after aneurysmal subarachnoid hemorrhage - a biphasic phenomenon. Clin Neurol Neurosurg. 2015 Oct;137:22-7. doi:10.1016/j.clineuro.2015.06.014. Epub 2015 Jun 19. PMID: 26123527.)

#. Response 1.

I agree with your concerns and fully recognize the importance of norepinephrine’s role. In our institution, the use of norepinephrine in SAH management was minimized with the primary goal of maintaining euvolemia and normotension. While we are aware that nimodipine, the calcium channel blocker (CCB) used based on evidence for vasospasm prevention in SAH patients, has minimal peripheral effects, administration of norepinephrine was strictly limited to cases where normotension could not be maintained.

Similar to the findings in the paper you suggested, we acknowledge that in most cases, normotension or even hypertension was maintained with the need for additional CCBs or ARBs. However, we would like to emphasize that this study does not primarily aim to assess whether pharmacological intervention is required to maintain blood pressure within a specific range to prevent vasospasm. Instead, this retrospective study evaluates whether ARBs have an impact on vasospasm prevention under euvolemic and normotensive conditions by considering their potential effects on "up or down regulations of the renin–angiotensin–aldosterone (RAA) system and the endothelin (ET) system, which are involved in the pathophysiological mechanisms" of cerebral vasospasm.

  1. CCB was added to ARB if needed – thus this is ARB-based therapy and not ARB alone or CCB alone, and ARB+CCB comprised a small proportion

#. Response 2.

I fully understand your concerns.

All patients were routinely administered nimodipine, and no additional medication was given to those who had no prior history of ARB use and were able to maintain euvolemia and normotension. Among patients without a history of ARB use, those who exhibited hypertension were not preferentially treated with ARBs; instead, the attending physician determined whether to administer an additional CCB or ARB. The difference in the proportion of patients receiving specific medications is one of the inherent limitations of a retrospective study.

  1. It is unclear if ARB(+/-CCB) were given before or after onset of vasospasm, before or after intervention – it appears to be ‘anytime’

#. Response 3.

Thank you for your comment

ARB was administered along with nimodipine at the initial stage based on the patient's medication history. Additional CCBs were given in cases where normotension was not maintained and persistent hypertension was observed. (Line 103-105)

  1. The achieved BP was not mentioned

#. Response 4.

Thank you for your comment.

The relevant information is provided in Line 108. However, there was an error in the recorded details, which has been corrected.

Postoperative systolic blood pressure was maintained within the range of 100-140 mmHg.

  1. This was not a randomised controlled trial, but rather an observational study.

#. Response 5.

Thank you for your comment.

This study is a retrospective observational study; therefore, it has certain limitations. These limitations have been described in the Limitation section (Lines 362-376).

  1. The way vasospasm was diagnosed is in doubt

#. Response 6.

Thank you for your comment.

Various methods for diagnosing vasospasm have been studied, and the criteria for vasospasm also vary significantly across different studies. In this study, we established diagnostic criteria based on previously reported research (Lines 290-302). We believe that our study provides additional evidence for vasospasm research and holds value in this context.

  1. That functional outcome (which is the end product of all processes including vasospam) was WORSE in the ARB group and this was glossed over by the authors

#. Response 7.

Thank you for your sharp insight.

You mentioned that we overlooked the poorer functional outcomes in the ARB group; however, we thoroughly discussed this issue within our study. We did not attribute the patient's functional recovery solely to vasospasm. Other factors, such as the patient’s initial condition upon admission, unexpected complications during treatment, and pre-existing comorbidities, could have influenced the outcomes. Most importantly, the small size of the ARB group may have also played a role, which cannot be ignored. We have addressed these limitations in the manuscript (Lines 369-374).

Nevertheless, the impact of ARBs on moderate to severe vasospasm was found to be statistically significant, and we believe that this study holds meaningful value in this regard.

The other issues:

  1. Line 2 – ‘for’ may be better replaced with ‘of’

#. Response 1.

Thank you for your comment.

As you pointed out, we have made the necessary corrections. (Line 2)

  1. Line 3 - to replace ‘after’ with ‘among patients who received interventions for’ - to better reflect the inclusion criteria

#. Response 2.

Thank you for your comment.

As you pointed out, we have made the necessary corrections. (Lines 3-4)

  1. Line 16 – why is ‘verify’ used? Are there prior reports of successful ARB use in humans with aSAH that need verification? Maybe say ‘investigate’ if there are no prior reports

#. Response 3.

Thank you for your comment.

As you pointed out, we have made the necessary corrections. (Line 20)

  1. Line 18 – Methods – briefly mention inclusion criteria

#. Response 4.

Thank you for your comment.

As you pointed out, we have made the necessary corrections. (Lines 23-24)

  1. Line 19 – clarify who these 210 patients are eg ‘with aSAH’

#. Response 5.

Thank you for your comment.

As you pointed out, we have made the necessary corrections. (Lines 23-34)

  1. Line 21 – Results - mention patient demographics and relevant characteristics

#. Response 6.

Thank you for your comment.

 As you pointed out, we have made the necessary corrections. (Lines 26-27)

  1. Lins 24-27 – better to mention the direction of the association eg increased or decreased

#. Response 7.

Thank you for your comment.

As you pointed out, we have made the necessary corrections. (Lines 30-33)

  1. Lines 29-31- may be better replaced as they do not attend to the study question. Eg say “Multicentre randomised blinded controlled trails are needed to further study the efficacy and safety of ARBs for the prevention of moderate to severe vasospasm among aSAH patients who had undergone intervention and had acute high blood pressure’

#. Resoponse 8.

Thank you for your advice.

We have revised the manuscript accordingly. (Lines 35-37)

  1. Line 42 – add ‘poor’ before ‘neurological’ to show the direction of the relationship

#. Response 9.

Thank you for your advice.

We have revised the manuscript accordingly. (Line 48)

  1. Line 46 – I understand nimodipine is used to reduce the risk of DCI and it has little material effect on the prevention and/or treatment of vasospasm – please quote the relevant reference if otherwise.

#. Response 10.

Thank you for your advice.

We have searched for relevant references; however, we could not find any literature opposing the effects of nimodipine. We did find studies discussing the effects of nimodipine in the context of traumatic SAH.

Effect of nimodipine on outcome in patients with traumatic subarachnoid haemorrhage: a systematic review.
Vergouwen MD, Vermeulen M, Roos YB. Lancet Neurol. 2006 Dec;5(12):1029-32. doi: 10.1016/S1474-4422(06)70582-8.

However, we were unable to find similar studies specifically related to aneurysmal SAH.

  1. Line 58 – ref 11 has been misquoted – I ask the authors to carefully read that paper.

#. Response 11.

We have carefully reviewed your comments.

The following content in the Results section of the referenced paper was cited as it discusses the findings of animal studies and preclinical research on ARBs. If you believe the reference is not appropriate, we are open to revising it accordingly. (Line 64)

The Role of Sartans in the Treatment of Stroke and Subarachnoid Hemorrhage: A Narrative Review of Preclinical and Clinical Studies. Brain Sci 2020, 10, doi:10.3390/brainsci10030153.

3.4. Effects of Losartan Following aSAH

LS, an already well-established antihypertensive drug in daily clinical practice and well examined in preclinical and clinical settings of ischemic stroke, shows promising results by attenuating cerebral inflammation and restoring cerebral autoregulation [64,105,122,123,124,125]. Facing preclinical aSAH research, beneficial effects of Sartans have been shown. Under already physiological conditions, LS diminished cerebral inflammation and associated DCVS [126] as well as ET-1 mediated vasoconstriction. Targeted ETB1- and ETA-R-antagonism under LS administration revealed a direct modulatory ETB1-R dependent effect via inducing upregulation of the NO-pathway with a significantly increased relaxation accompanied with enhanced sensitivity of the ETB1-R [23]. After induction of aSAH, ET-1-induced vasoconstriction was likewise decreased by LS preincubation, abolished after pretreatment with an ETB1-R antagonist. In precontracted vessels with LS and ETA-R-antagonism, ET-1 induced a higher vasorelaxation compared to the control group without, clearly demonstrating a modulatory and functional restoring effect of LS on the normally after aSAH impaired ETB1-R function [127].

Beneficial effects of LS on ET-1- and PGF2α-mediated DCVS after aSAH in a rat model have been reported, too [23,127]. An ET-1 mediated vasoconstriction was diminished, and ETB1-R mediated vasorelaxation under selective ETA-R blockade was restored [126,127]. In addition, PGF2α-elicited vasoconstriction of a basilar artery was markedly diminished [23,126,127]. Interestingly, several work groups could also verify positive vasomodulating effects of LS on the cerebral vessel wall, especially affecting SMCs [128,129]. Furthermore, aneurysm rupture was prevented in mice under LS treatment [129]. As already mentioned, after aSAH, increased synthesis of ET-1 triggers enhanced cerebral vasoconstriction; loss of the ETB1-R mediated vasorelaxation contributes to this effect, too [127]. Furthermore, upregulated AT2-1-R and PGF2α-synthesis contribute in enhancing and maintaining cerebral vasocontraction [7,130,131,132,133]. LS showed promising aspects in preclinical aSAH studies and therefore might have an effect in the treatment of patients with aSAH.

  1. Line 57 (major) – if CCBs and ARBs have actually been shown to be effective, what is the knowledge gap this study is trying to fill?

#. Response 12.

Thank you for your comments.

We identified an error in the sentence and have corrected it. While animal and preclinical studies have been reported, no clinical studies have been published. The necessary modifications have been made. (Line 64)

  1. Line 59 – is the aim to ‘verify’ what is already known? (See 2. above) What would be the added value of this study?

#. Response 13.

Thank you for your comment.

We corrected a word choice error, changing "verify" to "investigate." This study was conducted because, while animal and preclinical research exist, there have been no clinical studies. (Lines 64-65)

  1. Line 75 – please clarify that the ARB was received at any time during the admission (ie it can be before or after intervention, vasospasm)

#. Response 14.

Thank you for your comment.

The relevant content has been added and revised. (Lines 103-105)

  1. Lines 80 and 81 (major) – ‘post-treatment’ - is it ‘post-intervention’? ‘Post-admission? Or ‘post-onset of aSAH’ – though it is likely patients will be admitted on day of onset due to severe headache/loss of consciousness? Note that timing of vasospasms is usually based on day of onset

#. Response 15.

Thank you for your comment.

The necessary modifications have been made. (Line 86)

  1. Line 83 – ‘admission’ – do the authors mean pre-morbid? ‘discharge’ - mRS at 3 months post-aSAH may be better measure of functional outcome than at discharge

#. Response 16.

Thank you for your comment.

In our hospital, patients with minimal damage are discharged early, whereas those with neurological abnormalities undergo sufficient rehabilitation and treatment before discharge, resulting in no significant difference in the mRS at 3 months. Since the discharge period was not uniform, we initially used "discharge", but we have now revised it to "mRS at 3 months." (Line 89)

  1. Lines 86, 128, 135 – and throughout the text where relevant (major) - ‘treatment may be better replaced with ‘intervention’

#. Response 17.

Thank you for your comments.

The necessary revisions have been made. (Lines 90, 135, 142)

  1. Line 96 – what is the target BP range for ‘normotension’? What ARBs and CCBs were used?

#. Response 18.

Thank you for your comment.

There was an error in the writing process, which has now been corrected.

We have specified the target range for normotension. (Line 108)

The ARB used was losartan, and the CCB used was nicardipine. This information has been added. (Line 102)

  1. Section 2.3 and 4.1 (very major) – it is unclear what was used to diagnose moderate to severe vasospasm? TCD, MRA? TFCA? Any of them? If any, the authors need to show that all the 3 methods used for diagnosing vasospam are highly correlated

#. Response 19.

Thank you for your comments.

The relevant content has been added. (Lines 144-151)

The correlation analysis has been discussed in the Discussion section. (Lines 276-289)

  1. Line 131 (major) – I have concerned that only velocities were used to diagnose vasospasm when the recommendation is to use the Liebegard (MVA/tICA) ratio as well (Alexandrov AV, Sloan MA, Wong LK, Douville C, Razumovsky AY, Koroshetz WJ, Kaps M, Tegeler CH; American Society of Neuroimaging Practice Guidelines Committee. Practice standards for transcranial Doppler ultrasound: part I—test performance. J Neuroimaging. 2007 Jan;17(1):11-8. doi:10.1111/j.1552-6569.2006.00088.x. PMID: 17238867.)

#. Response 20.

Thank you for your comment.

We also agree with your point and have provided further clarification on this matter. (Lines 144-151)

  1. Lines 136-137 (major) – what is the authoritative reference for this angiographic criteria?

#. Response 21.

Thank you for your comment.

In the Discussion section (4.1. Diagnostic modalities and criteria of cerebral vasospasm), we referenced studies that have diagnosed moderate vasospasm based on a 25% to 33% or more reduction in lumen diameter using angiographic imaging [4,5,16,21].

Based on this information (Lines 294-296), we classified moderate to severe vasospasm as a ≥33% reduction in lumen diameter.

  1. Line 137 - what is the authoritative reference for this DCI criteria?

#. Response 22.

Thank you for your comments.

Reference 4 is the relevant reference, and we have included it accordingly. (Lines 151-154)

  1. Table 1 – please check the p-value for HTN, unlikely to be 0.941. M1/ICA ratio on which test? What is ‘pre-mRS’ – pre-morbid? On admission pre-intervention? If there was less vasospasm in ARB group, why are the TCD velocities not different – even more relevant issue as table 2 shows differences(major)

#. Response 23.

Thank you for your comments.

There was an error in the recorded information, and we have corrected the P-value for HTN accordingly. (Lines 195 and 212)

At the request of Reviewer #2, we modified the statistical methods and updated the manuscript accordingly. (Line 212)

The M1/ICA ratio was primarily assessed using TFCA when available. However, in cases where TFCA was not performed or could not be conducted, MRA was used for analysis.

Pre-mRS has been corrected and recorded as pre-procedure in Line 89.

In Table 1, while the ARB group had a lower incidence of vasospasm, the absolute number of vasospasm cases in the ARB group was very small, which likely limited its impact on the mean values. Although the TCD values were numerically lower in the ARB group, the difference was not statistically significant.

Table 2 presents the differences in TCD values between patients with and without vasospasm, rather than comparing the ARB and no ARB groups. Therefore, its findings should be interpreted separately.

  1. Line 200 - is ‘operation’ different from ‘intervention’?

#. Response 24.

Thank you for your comments.

"Intervention" has been revised accordingly. (Line 221)

  1. Table 2 – do not use ‘p=0.000’, instead provide a value eg <0.001 if that is correct

#. Response 25.

Thank you for your comments.

The necessary revisions have been made. (Lines 223-224)

  1. Results (major) – 4.1 - this should be spent discussing the results of their internal comparison of 3 methods the authors used, and the lack of congruity as mentioned in 23. Above

#. Response 26.

Thank you for your comments.

The relevant content has been added. (Lines 304-309)

As mentioned above, TCD, MRA, and TFCA are all essential methods for effectively diagnosing vasospasm. In this study, although the incidence of vasospasm was statistically lower in the ARB group, the TCD values did not show a significant statistical difference. This may be due to the overall low number of vasospasm cases in both groups, which could have influenced the results. To address this limitation, a large-scale prospective randomized study would be necessary.

Nevertheless, this study holds significance as it evaluates moderate to severe vasospasm. (Lines 304-309)

  1. Table 3 - what about the detrimental effects of norepinephrine on vasospasm? (Zeiler FA, Silvaggio J, Kaufmann AM, Gillman LM, West M. Norepinephrine as a potential aggravator of symptomatic cerebral vasospasm: two cases and argument for milrinone therapy. Case Rep Crit Care. 2014;2014:630970. doi:10.1155/2014/630970. Epub 2014 Nov 9. PMID: 25431686; PMCID: PMC4241707.)

#. Response 27.

Thank you for your comments

Regarding our hospital’s post-interventional management, norepinephrine was used in a very limited manner, to the extent that statistical analysis was not feasible.

As shown in Table 1, the ARB group primarily consisted of patients who already had hypertension (HTN), and due to the elevated blood pressure caused by hemorrhage, it would be difficult to assess the impact of norepinephrine in this study. We kindly ask you to consider this point.

  1. Line 294-296 – the direction of the association should be stated

#. Response 28.

Thank you for your comments

Revised and recorded accordingly. (Line 321-323)

  1. Line 305 (major) – a detailed explanation of this finding needs to be done as it is counter to the literature

#. Response 29.

Thank you for your comments

Relevant content has been added. (Line 333-336)

  1. Line 327 – again, ‘verify’ may not be appropriate

#. Response 30.

Thank you for your comments

The necessary revisions have been made and recorded accordingly. (Line 358)

  1. Limitations – the authors should add that the allocation d ARB was not randomised

#. Response 31.

Necessary revisions have been made as per your comment. (Line 375-376)

Reviewer 2 Report

Comments and Suggestions for Authors

- Specify in the title or in the Materials and Methods section that it is a retrospective study.

- It would be useful to define the primary and secondary endpoints in the abstract and in the Materials and Methods and Results sections.

- In the introduction, it is explained that the incidence of vasospasm, DCI, etc. following subarachnoid hemorrhage is variable; however, it would be more complete to specify this incidence in percentage terms to make these statements objective.

- Which study is being referred to in lines 44-47?

- What are the current indications for the use of ARBs, and why were they not administered to patients in group B?

- For better understanding, it would be useful to describe in the Materials and Methods section what treatment was administered to group A and what to group B to understand if they actually differ only in the administration of ARBs or if there are confounding factors.

- For retrospective studies, it is useful to illustrate OS (Overall Survival) and event-free survival with Kaplan-Meier curves complete with standard error and number of patients at risk.

- Given the numerical disparity in the two groups, it would be useful to add confidence intervals for the examined variables and regression analysis.

Author Response

Response to Reviewer’s comment

Thank you very much for taking the time to review this manuscript. Please find the detailed responses below and the corresponding revisions/corrections highlighted in the re-submitted files.

Reviewer #2.

  1. Specify in the title or in the Materials and Methods section that it is a retrospective study.

#. Response 1.

Thank you for your comment.

We have already stated that this study was conducted as a retrospective analysis in the Materials and Methods section (Line 72). Please confirm this.

  1. It would be useful to define the primary and secondary endpoints in the abstract and in the Materials and Methods and Results sections.

#. Response 2.

Thank you for your comment.

The primary endpoint of this study is the incidence of moderate to severe vasospasm. Moderate to severe vasospasm was assessed by dividing patients into the ARB group and no ARB group.

The secondary endpoint is to identify independent risk factors associated with moderate to severe vasospasm. (Lines 156-160)

  1. In the introduction, it is explained that the incidence of vasospasm, DCI, etc. following subarachnoid hemorrhage is variable; however, it would be more complete to specify this incidence in percentage terms to make these statements objective.

#. Response 3.

Thank you for your comment.

Following your advice, the incidence has been explicitly stated as a percentage. (Line 43)

  1. Which study is being referred to in lines 44-47?

#. Response 4.

Thank you for your comment.

The reference has been added accordingly. (Line 53)

  1. What are the current indications for the use of ARBs, and why were they not administered to patients in group B?

#. Response 5.

Thank you for your comment.

The indications for ARB use have been documented in the 2.2. Patient Management section (Lines 103-105).

ARB was administered to patients with a prior history of ARB use based on their medical history.

  1. For better understanding, it would be useful to describe in the Materials and Methods section what treatment was administered to group A and what to group B to understand if they actually differ only in the administration of ARBs or if there are confounding factors.

#. Response 6.

Thank you for your comment.

The management of the ARB group and the no ARB group was conducted in the same manner without any differences.

As a limitation of a retrospective study, treatment methods could not be predetermined or standardized. Instead, statistical analysis was performed to compare differences between the ARB and no ARB groups.

However, using multivariate analysis, we were able to identify independent factors associated with moderate to severe vasospasm.

  1. For retrospective studies, it is useful to illustrate OS (Overall Survival) and event-free survival with Kaplan-Meier curves complete with standard error and number of patients at risk.

#. Response 7.

Thank you for your comment.

The occurrence of vasospasm was observed around day 7 and, in all cases, within two weeks. As a result, the Kaplan-Meier curve did not provide meaningful findings.

Additionally, when comparing the ARB group and the no ARB group, there was no statistically significant difference between the two groups. Therefore, we determined that presenting the Kaplan-Meier curve as a result would be inappropriate.

  1. Given the numerical disparity in the two groups, it would be useful to add confidence intervals for the examined variables and regression analysis.

#. Response 8.

We appreciate your valuable feedback.

We have inserted the confidence interval in Table 1. (Lines 166-167 and 212)

Round 2

Reviewer 1 Report

Comments and Suggestions for Authors

This paper is a revised submission of a report on a single-centre retrospective cohort study of the frequency of moderate to severe vasospasm among patients with aneurysmal subarachnoid haemorrhage (aSAH) depending on whether they had received angiotensin-receptor blocker (ARB).

I appreciate that the authors have addressed many of my concerns. However, a few minor ones remain:

1.      In their Response #1, and lines 53-54 the authors maintain that ‘nimodipine, the calcium channel blocker (CCB) used based on evidence for vasospasm prevention in SAH patients, has minimal peripheral effects’ - this is untrue. Nimodipine does not primarily prevent vasospasm but rather the effects of delayed cerebral ischaemia, it also can cause hypotension especially if administered intravenously (Nimotop product monograph from Bayer https://docisolation.prod.fire.glass/?guid=0381c45a-814c-4226-fd98-f586da4af4f0)

2.      Lines 26-27  - ‘patients without vascular abnormality’ may not be correct as they likely have aneurysms?

3.      Line 27 – as mentioned before, please provide more demographic data eg mean and SD or median with IQR for age, sex ratio

4.      Lines 53-54 – as mentioned above, nimodipine does not primarily prevent vasospasm but rather the effects of delayed cerebral ischaemia, it also can cause hypotension especially if administered intravenously (Nimotop product monograph from Bayer https://docisolation.prod.fire.glass/?guid=0381c45a-814c-4226-fd98-f586da4af4f0)

5.      Line 105 – to use ’intervention’ instead of treatment?

6.      Line 107 – please add an explanation for what was done for patients without a history of ARB administration

Author Response

Comments and Suggestions for Authors

Reviewer #1.

This paper is a revised submission of a report on a single-centre retrospective cohort study of the frequency of moderate to severe vasospasm among patients with aneurysmal subarachnoid haemorrhage (aSAH) depending on whether they had received angiotensin-receptor blocker (ARB).

I appreciate that the authors have addressed many of my concerns. However, a few minor ones remain:

Response #0.

We sincerely appreciate your insightful and meticulous questions, as well as your thorough and critical review of our manuscript. Your detailed assessment and constructive feedback have been invaluable in improving the clarity and quality of our study.

  1. In their Response #1, and lines 53-54 the authors maintain that ‘nimodipine, the calcium channel blocker (CCB) used based on evidence for vasospasm prevention in SAH patients, has minimal peripheral effects’ - this is untrue. Nimodipine does not primarily prevent vasospasm but rather the effects of delayed cerebral ischaemia, it also can cause hypotension especially if administered intravenously (Nimotop product monograph from Bayer https://docisolation.prod.fire.glass/?guid=0381c45a-814c-4226-fd98-f586da4af4f0)

Response #1. Thank you for your careful and detailed feedback.

We believe there was an error in the wording. It would be more accurate to state that "Pharmacodynamically, nimodipine has a relatively greater effect on cerebral circulation than on peripheral circulation.

Please refer to the following information.

"Reference for peripheral effect of nimodipine”

While nimodipine can cause some peripheral vasodilation, leading to a slight drop in blood pressure, this effect is usually much less pronounced compared to its impact on cerebral vessels. 

Nimodipine: a new calcium antagonistic drug with a preferential cerebrovascular action

2025 Feb;117(2):589-597.  doi: 10.1002/cpt.3499.

Cerebral Ischemia Protection After Aneurysmal Subarachnoid Hemorrhage: CSF Nimodipine Levels After Intravenous Versus Oral Nimodipine Administration

2025 Feb;117(2):589-597.  doi: 10.1002/cpt.3499. Epub 2024 Nov 19.

I agree with your opinion. IV nimodipine may cause a decrease in blood pressure; however, if an appropriate dose is administered while maintaining normotension, the synergistic effect of nimodipine’s benefits—prevention and treatment of vasospasm and ischemic-related complications—combined with the effect of ARBs on vasospasm can be maximized.

There are reported references on the possible mechanisms of nimodipine's preventive effect on vasospasm. We have included these reference and revised the manuscript accordingly (Line55-57, Line 445-447)

Beyond nimodipine: advanced neuroprotection strategies for aneurysmal subarachnoid hemorrhage vasospasm and delayed cerebral ischemia

Neurosurgical Review (2024) 47:305 https://doi.org/10.1007/s10143-024-02543-5

Nimodipine pharmacodynamic, pharmacokinetic, strengths and weaknesses The incidence of vasospasm in large series has been reported to be higher in medial and anterior circulation aneurysms [130–134], while lower rates have been observed in ruptured posterior cerebral artery aneurysms [135]. These findings have also been confirmed in the authors’ own series [136–145]. Nimodipine is the backbone in the prevention and treatment of vasospasm and ischemic related complications [11, 146, 147], it being the only medication approved by the US Food and Drug Administration and recommended by the AHA/ASA guidelines [12]. Although it is known to inhibit the influx of calcium ions through voltage-gated L-type calcium channels of vascular smooth muscles [148], this phenomenon is not held responsible for improving clinical outcome of SHA and DCI patients. Possible proposed mechanisms of action targeting vasospasm are instead increase in fibrinolytic activity, neuroprotection, and inhibition of cortical spreading ischemia [149–151].

Please review the content with the provided references. If you believe our suggestions are not accurate, we will gladly re-examine and make any necessary corrections.

  1. Lines 26-27  - ‘patients without vascular abnormality’ may not be correct as they likely have aneurysms?

Response#2.

Thank you for your comment.

The suggested revisions have been made and incorporated accordingly.

A total of 181 patients aged 19 years or older with aSAH, without vascular abnormalities (including vascular malformations and moyamoya disease), were enrolled in this study. (Line26-28)

  1. Line 27 – as mentioned before, please provide more demographic data eg mean and SD or median with IQR for age, sex ratio.

Response#3.

Thank you for your comment.

The suggested revisions have been made and incorporated accordingly.

The age of the enrolled patients was 59.01 ± 12.985 (mean ± standard deviation), and the sex ratio of males to females was 66:115, with a higher proportion of females (Line 28-30)

  1. Lines 53-54 – as mentioned above, nimodipine does not primarily prevent vasospasm but rather the effects of delayed cerebral ischaemia, it also can cause hypotension especially if administered intravenously (Nimotop product monograph from Bayer https://docisolation.prod.fire.glass/?guid=0381c45a-814c-4226-fd98-f586da4af4f0)

#. Response #4

Thank you for your feedback.

I agree with your opinion.

As stated in the response to Comment #1, IV nimodipine may cause a decrease in blood pressure. However, if an appropriate dose is administered while maintaining normotension, the synergistic effect of nimodipine’s benefits—prevention and treatment of vasospasm and ischemic-related complications—combined with the effect of ARBs on vasospasm can be maximized.

  1. Line 105 – to use ’intervention’ instead of treatment?

#. Response #5

Thank you for your comment.

The content has been revised and updated accordingly. (Line 107)

  1. Line 107 – please add an explanation for what was done for patients without a history of ARB administration

#. Response #6

Thank you for your comment.

The revisions have been made and incorporated accordingly.

For patients without a history of ARB administration, only nimodipine was initially administered after the intervention. If blood pressure was not adequately controlled, the attending physician had the discretion to administer either an ARB or a CCB, or to prescribe both ARB and CCB together. (Line 109-112)

Reviewer 2 Report

Comments and Suggestions for Authors

Thank you for corrections and reponses.

1. It's important to mention that the study is retrospective in the abstract, even if this information is already present in the Materials and Methods section. Many readers of scientific articles, especially those doing a quick literature search, only read the abstract to decide if the article is of interest. If they don't find the key information about the study type (retrospective in this case) in the abstract, they may not read further and miss important information. The abstract should be a concise but complete summary of the study. Repeating that it is a retrospective study helps to clarify the study design right away, without forcing the reader to look for this information in the text. It's an editorial standard, many scientific journals require that the study type (retrospective, prospective, randomized, etc.) be specified in both the abstract and the Materials and Methods section.

7. In a retrospective study, it is always useful to express events in relation to time, even if the time span in which the study was conducted is not large. Even if no statistically significant differences between groups are found in a retrospective study, Kaplan-Meier curves remain a valuable tool for visualizing the temporal distribution of events, comparing results with other studies, and identifying potentially interesting trends. However, it is not essential.

Author Response

Comments and Suggestions for Authors

Reviewer #2.

Thank you for corrections and responses.

Once again, I would like to express my deepest gratitude for your thorough and thoughtful review.

  1. It's important to mention that the study is retrospective in the abstract, even if this information is already present in the Materials and Methods section. Many readers of scientific articles, especially those doing a quick literature search, only read the abstract to decide if the article is of interest. If they don't find the key information about the study type (retrospective in this case) in the abstract, they may not read further and miss important information. The abstract should be a concise but complete summary of the study. Repeating that it is a retrospective study helps to clarify the study design right away, without forcing the reader to look for this information in the text. It's an editorial standard, many scientific journals require that the study type (retrospective, prospective, randomized, etc.) be specified in both the abstract and the Materials and Methods section.

Response #1

Thank you for your advice.
The content has been incorporated accordingly. (Line20)

  1. In a retrospective study, it is always useful to express events in relation to time, even if the time span in which the study was conducted is not large. Even if no statistically significant differences between groups are found in a retrospective study, Kaplan-Meier curves remain a valuable tool for visualizing the temporal distribution of events, comparing results with other studies, and identifying potentially interesting trends. However, it is not essential.

Response #7

Thank you for your advice.

The analysis has been included along with the corresponding figure.

The vasospasm-free survival rate according to the ARB administration was compared using the Kaplan-Meier method and statistical differences were revealed by log-rank test (Line 173-175)

There was no statistically significant difference in the moderate to severe vasospasm-free survival rate between the no-ARB group and the ARB group (P=0.699) (Figure 2). (Line 219-221)

Figure 2. Kaplan-Meier curve depicting the vasospasm-free survival rate based on ARB administration (Line227-229)
